# Modelling and Measurement of Magnetically Soft Nanowire Arrays for Sensor Applications

**DOI:** 10.3390/s21010003

**Published:** 2020-12-22

**Authors:** Pavel Ripka, Vaclav Grim, Mehran Mirzaei, Diana Hrakova, Janis Uhrig, Florian Emmerich, Christiane Thielemann, Jiri Hejtmanek, Ondrej Kaman, Roman Tesar

**Affiliations:** 1Faculty of Electrical Engineering, Czech Technical University in Prague, Technicka 2, 166 27 Praha 6, Czech Republic; vaclav.grim@fel.cvut.cz (V.G.); mirzameh@fel.cvut.cz (M.M.); hrakodia@fel.cvut.cz (D.H.); 2Biomems Lab, Faculty of Engineering, Technische Hochschule Aschaffenburg, 63743 Aschaffenburg, Germany; s130567@h-ab.de (J.U.); florian.emmerich@h-ab.de (F.E.); christiane.thielemann@h-ab.de (C.T.); 3Fyzikální Ústav AV ČR, v. v. i, Cukrovarnicka 10/112, 162 00 Praha 6, Czech Republic; hejtman@fzu.cz (J.H.); kaman@fzu.cz (O.K.); tesar@fzu.cz (R.T.)

**Keywords:** magnetic nanowires, soft magnetic wires, magnetic sensors

## Abstract

Soft magnetic wires and microwires are currently used for the cores of magnetic sensors. Due to their low demagnetization, they contribute to the high sensitivity and the high spatial resolution of fluxgates, Giant Magnetoimpedance (GMI), and inductive sensors. The arrays of nanowires can be prepared by electrodeposition into predefined pores of a nanoporous polycarbonate membrane. While high coercivity arrays with square loops are convenient for information storage and for bistable sensors such as proximity switches, low coercivity cores are needed for linear sensors. We show that coercivity can be controlled by the geometry of the array: increasing the diameter of nanowires (20 µm in length) from 30 nm to 200 nm reduced the coercivity by a factor of 10, while the corresponding decrease in the apparent permeability was only 5-fold. Finite element simulation of nanowire arrays is important for sensor development, but it is computationally demanding. While an array of 2000 wires can be still modelled in 3D, this is impossible for real arrays containing millions of wires. We have developed an equivalent 2D model, which allows us to solve these large arrays with acceptable accuracy. Using this tool, we have shown that as a core of magnetic sensors, nanowires are efficiently employed only together with microcoils with diameter comparable to the nanowire length.

## 1. Introduction

Magnetic wires have been used as functional materials of fluxgate magnetic field sensors since the 1930s, and some of these devices are still in production. The magnetic core of these sensors is typically a permalloy (NiFe) wire with a diameter of 0.2 mm, but some fluxgates and induction sensors use core in the form of rod with length up to 1 m [1]. While wire-core fluxgates usually do not achieve the low noise and the high offset stability of ring-core fluxgate sensors made of thin tape, wire cores of Vacquier type show several fundamental advantages [2]:The sensing direction is defined by the direction of the sensor core and not by the direction of the pickup coil. This allows the construction of highly stable gradiometers of the Foerster type [3].The wire-core sensor has high spatial resolution, and it is therefore convenient for measurements of small field sources such as microbeads [4,5].The high shape anisotropy reduces the Crossfield error (a non-linearity response to fields perpendicular to the primary sensing direction) [6].The low demagnetization of the sensor core increases sensitivity and thus allows miniaturization of sensors.

Wires have low cross-sectional area, but the decreased sensitivity can be compensated by increasing of the excitation frequency, as the effect of eddy currents is less pronounced [7]. Magnetic wires have been also used for other sensors such as Wiegand sensors used for position detectors, speed sensors, security labels and for energy harvesting [8].

A new era of magnetic wires started in 1976 with the invention of amorphous microwires. These wires, typically 50 μm to 150 μm in diameter, are mechanically strong and exhibit magnetically soft properties even without annealing [9,10]. They are therefore well suited for applications in fluxgate and GMI magnetic field sensors, and for strain measurements [11]. The magnetic properties of these wires depend on their chemical composition and cooling rate; they can be further tailored by field annealing or by stress annealing. Helical anisotropy can be established by annealing under torsion [12], allowing for the construction of coil-less fluxgate sensors [13,14], whereas other sensors use domain-wall velocity [15]. Passive wireless strain sensors based on microwires can be embedded into composite structures [16]. Glass-covered wires are produced by the Tailor method, which allows the production of diameters from below 1 μm up to 50 μm [17]. In as-cast form, these wires suffer from internal stresses causing non-repeatability and increased noise level. These stresses can be released by current annealing [18]. Both crystalline structures and amorphous structures can be achieved by this technique. Amorphous wires can be subsequently nanocrystallized by thermal treatment [19].

The first sensor application of amorphous wire was the GMI sensor [20,21], where the high-frequency impedance of the wires depends on the DC field that is applied, due to a change in the magnetic permeability. GMI sensors can detect fields down to the nT range, noise level of 35 pT/√Hz at 1 Hz has been recently reported [22]. GMI sensors based on amorphous wire have reversible and reproducible stress sensitivity [23]. The main problem of GMI sensor is their poor DC offset stability, caused by the fact that their magnetic core is not saturated. Thus, the sensor can be magnetized by a strong magnetic field, which causes an offset shift. It is also difficult to stabilize the bias field which is necessary to achieve linear operation [24]. Despite these disadvantages, GMI sensors are being used for the detection on nanoparticles [25] and integrated GMI sensors are used in mobile phones [26].

A promising application could be the use of a microwire as a core for a miniature fluxgate sensor, which can even be flexible [27,28]. Fluxgate sensors utilize non-linearity of the magnetization characteristics of the soft magnetic core. In the presence of the measured DC field, the characteristics shifts, and even harmonic components of the excitation frequency appear in the induced voltage. The output signal asymmetry can also be detected in the time domain [29,30]. Fluxgates are usually excited by a strong AC field, which magnetizes the core deeply into saturation during each excitation cycle. Due to this fact, the magnetic state of the core is restored, and the sensor has a DC stability of typically 1 nT. This is true for the longitudinal fluxgate, for which the wire core is excited in the longitudinal direction by a solenoid coil [31]. By contrast, the transverse fluxgate is excited by an electric current through the wire. In order to fully saturate the magnetic core, composite wires consisting of a copper core and a ferromagnetic shell were fabricated by electrodeposition [32]. Microwires are also used as security labels and in microwave metamaterials [33]. Magnetic microwires have low coercivity, high permeability, and may achieve near-zero magnetostriction. In this regard they are superior to thin-film cores which are manufactured by sputtering or electrodeposition. The mentioned properties make microwires ideal for microfluxgate sensors [34,35] which have applications in mobile devices, motion tracking, medical devices, non-destructive testing and in the mining industry [36]. A 20 mm long fluxgate sensor based on a single amorphous microwire achieved a noise of 1.4 nT/√Hz at 1 Hz [37]

In 2009, we studied transverse fluxgates with cores made of several microwires. We found that the performance is strongly affected by the magnetostatic coupling between the wires, which depends on their distance [38]. The demagnetization factor of the microwire core was studied in [39]. At that time, 3D Finite Element Modelling (FEM) of a multiwire core was computationally challenging, so that the study was limited to several wires only.

Nanowire arrays have been fabricated by electrodeposition of a magnetic material into the pores of nanoporous membranes [40]. These arrays exhibit ferromagnetic resonance in the GHz range, and they have large potential for microwave applications such as tunable filters, circulators, and nanoantennas up to THz range [41,42]. Individually functionalized nanowires can also be used in biomedical applications [43]. Another important application of nanowire arrays is in perpendicular magnetic recording [44]. Magnetic nanowires have been fabricated in several laboratories, but to the best of our knowledge they have never been used as functional materials for magnetic field sensors, as most of the fabricated wires have high coercivity. For CoNiP material system the minimum achieved coercivity was 64 kA/m [45]. A systematic study based on micromagnetic simulations of the remagnetization process for cylindrical nanowires with different crystalline structure was conducted in [46].

In this paper, we show progress that has been made in the fabrication, testing and modelling of magnetically soft nanowires. In detail, we examine the effect of the nanowire array geometry on the global magnetic properties. Finite element simulations of nanowire arrays are performed with a simplified 2D equivalent model, which allows for the modelling of large arrays, and the results are verified with the experimental data.

## 2. Nanowire Fabrication

Our magnetic nanowires are grown by electrodeposition of a magnetic permalloy (NiFe) into the pores of a nanoporous membrane. Here, we use polycarbonate (PC) membranes for the fabrication of nanowires of various diameters and lengths, but alumina and silicon membranes can be used as well.

The general production process of magnetic nanowire arrays is schematically depicted in Figure 1. At first, a glass substrate is metallized with a 200 nm thick copper layer by using a sputter process. In the next step, a photoresist SU8 (MicroChem Corp., Westborough, MA, USA) is structured on top to form an insulation layer, determining the nanowires growth area (Figure 1a). Subsequently, an ion track etched PC membrane (Merck KGaA, Darmstadt, Germany) is placed on the prepared substrate and is gently pressed with a sponge (Figure 1b). This is necessary to avoid any voids between the membrane and the prepared structure, which would result in irregular deposition. Controlled filling of the holes between substrate and membrane is performed by a galvanic growth process (Figure 1c). The electrochemical deposition of the permalloy is carried out at 35 °C in a three-electrode setup, where a platinum wire is used as a counter electrode and Ag/AgCl is used as a reference electrode. Two hours prior to deposition and during deposition, the electrolyte was bubbled with nitrogen to avoid oxidation of Fe^2+^ ions. During pulsed electrochemical deposition, the composition of the nanowires’ material is controlled by the parameters of the voltage pulses. This technique has been shown to achieve homogeneous growth of the wires [47]. The delay time between pulses ensures constant material transport through the pores and helps to renew the concentration of the metal ions at the pore-electrolyte interface. All deposition processes were carried out by using VersaStat4 (Princeton Applied Research, Oakridge, TN, USA), since this instrument allows for very precise adjustment of the required parameters. For the pulsed deposition, an aqueous electrolyte was developed which consists of the following components: 300 g/L NiSO_4_.6H_2_O, 45 g/L NiCl_4_.6H_2_O, 43.27 g/L FeSO_4_.7H_2_O, 45 g/L H_3_BO_3_ [48].

Boric acid was used to enhance the ion transport and the solubility of all components. During pulsed deposition, a constant deposition pulse time (t_depo_) of 10 ms and a delay time (t_delay_) of 100 ms were applied, with voltage amplitude of −1.2 V and −0.7 V, respectively (Figure 1f), which resulted in uniform nanowire growth. The morphology and the size of the nanowires were characterized by scanning electron microscopy (SEM) (Phenom ProX), and the micrographs are shown in Figure 2. The chemical composition of the wires was determined to Ni_80_Fe_20_ by energy dispersive X-ray spectroscopy (EDX) attached to the SEM. An EDX spectrum of the Ni_80_Fe_20_ nanowires grown in the PC template is shown in Figure 3 (see the red line).

In another experiment, we varied the composition of the nanowire material by decreasing the pulse delay time during electrochemical deposition from t_delay_ = 100 ms to t_delay_ = 50 ms, which resulted in a reduction in iron content of 10%, leading to Ni_90_Fe_10_), see Figure 3. In both samples, no signatures of oxygen contamination were found, as they were kept under a constant nitrogen atmosphere. The nanowire lengths were controlled by monitoring the current during deposition and adjusting the deposition time, as the length increased linearly with time. In addition to the nanowire arrays depicted in Figure 2, we fabricated nanowires with diameters of 30 nm and 400 nm by employing the same process. The nanowire length was always 20 µm. In summary, we have presented a method for fabrication of permalloy nanowires that allows the geometry and the composition of the nanowires to be controlled. The quality of the membrane is very important, as it determines the density and the geometry of the wires.

Finally, as an optional step, the membrane can be dissolved with acetone, which would release free-standing nanowires (Figure 1e). However, in this study we kept the nanowires inside the membrane to ensure that they were immovable.

## 3. Nanowire Magnetic Characterization

Magnetic measurements in DC fields at a temperature of 300 K were performed by SQUID magnetometry on a MPMS XL (Magnetic Property Measurement System, Quantum Design, Inc.). In order to evaluate the effect of demagnetization, we measured samples of two diameters, 1 mm and 3 mm, from each type of nanowire arrays. The measuring sequence was programmed to achieve high precision of the coercivity measurement and to keep the measurement time reasonably short: at first the magnet reset option (controlled quench) was applied to remove any remnant fields in the superconducting winding of the magnetometer, and a full magnetization loop was measured within the range ±159 kA/m (±200 mT) with no-overshoot approach and stabilization of particular fields during the field scan, close to zero field the linear regression mode for fitting SQUID scans was used instead of the iterative regression mode. After the measurement of these low-field loops, the magnetization curves were measured up to higher fields of 3180 kA/m (4T) to determine the saturation magnetization.

The complete hysteresis loop in the high-field range is shown on arrays with different wire diameters in Figure 4. Figure 5 shows examples of the detailed low-field hysteresis loop measured on 30 nm, 200 nm and 400 nm diameter wire arrays. Importantly, knowing the saturated magnetic moment of each sample and the saturated magnetic flux density Bs of the electrodeposited permalloy with known chemical composition, we can rescale the y axis in units of B (Figure 5b) and to calculate the apparent permeability of the measured nanowire array.

The measured coercivity values are shown in Table 1. The coercivity of samples A (30 nm diameter wires) occurs in the range ≈30–40 kA/m. Its value drops significantly with increasing the wire diameter, specifically to ≈5 kA/m and below for diameters of 200 nm and 400 nm. The coercivity does not show significant dependence on the sample size (a diameter of 1 mm or 3 mm). The magnetic moment per unit area is proportional to the amount of magnetic material present in the sample, which depends on the wire diameter. Sample-to-sample variations are probably caused by the random character of the pores in the polycarbonate membrane, as the chemical composition of the wires was quite stable. The apparent permeability was calculated as a slope of the BH curve. The values of flux density B were calculated from the measured magnetic moment, supposing that the saturated magnetic moment is always equivalent to B = 0.7 T, which is the saturated flux density of the permalloy that was employed in the present study.

The origin of the reduction in coercivity with growing the wire diameter can be explained by the increased magnetostatic coupling between wires together with an increase in demagnetization, which leads to the decrease in shape anisotropy. This is demonstrated by the hysteresis loop measured in the direction perpendicular to the wires (Figure 6). For the selected wire diameter of 200 nm, the coercivity in the perpendicular direction is Hc_┴_ = 8.5 kA/m, while the coercivity in the longitudinal direction is only Hc_║_ = 2.6 kA/m (Figure 5, Table 1). Also, the relative permeability in the perpendicular direction is μ_┴_ = 5.4 and in the longitudinal direction it is μ_║_ = 3.3, which indicates that the easy direction is already perpendicular to the wire axis.

## 4. Modelling Nanowire Arrays

### 4.1. Demagnetization, Apparent Permeability and Amplification Factor

The demagnetization factor of a single wire was calculated with high accuracy by Chen et al., assuming a constant permeability [49]. We verified his calculation by 3D FEM modelling, and we calculated more datapoints to improve the interpolation errors.

We also calculated the values of the (magnetometric) apparent permeability *μ*_a_, which is defined as:(1)$$\mu_{a}=\frac{B_{\mathrm{mean}}}{\mu_{0}H_{0}}$$
where B_mean_ is the average value of magnetic flux density within the wire volume that was inserted into the homogeneous field with intensity H_0_.

The relation between the magnetometric demagnetization factor D and apparent permeability is described by the formula [50]
(2)$$\mu_{a}=\frac{\mu_{r}}{1+D\left(\mu_{r}-1\right)}$$
where *μ*_r_ is relative permeability of the material.

While the apparent permeability of ring and racetrack cores have been extensively studied, we are not aware of any paper analyzing the effects of the coil geometry and core geometry on the sensitivity of a multicore sensor.

Verification of our calculations was performed by measurements on the array of crystalline permalloy microwires. The models should also be extended to include non-linear magnetization curves of the material under study. We defined the amplification factor *a* of the induction coil with a multiwire core:(3)$$a=\frac{\Phi_{\mathrm{cored}}}{\Phi_{\mathrm{air}}}$$
where Φ cored and Φ_air_ is the coil flux with and without the core, respectively.

For very slim coils wound tightly around a rod core, the amplification factor is roughly equal to the apparent permeability *μ*_a_; this is not valid for wire cores with large coil area.

Better approximation of the amplification factor *a* considers the coil cross-sectional area A*_air_* and core cross-section A*_core_*. This formula was derived by Primdahl for fluxgate sensors and it is commonly used in literature [51]:(4)$$a=\frac{\Phi_{cored}}{\Phi_{air}}=\frac{A_{air}-A_{w}+\mu_{a}A_{w}}{A_{air}}=1+(\mu_{a}-1)\frac{A_{w}}{A_{air}}$$
where A*_w_*/A*_air_* is array density.

We have already shown by FEM simulations and verified by measurement that for small wire arrays the real values of the amplification factor are much lower [50].

In this paper we examine the apparent permeability and amplification factor of nanowire arrays as a function of distance between the wires, i.e., wire density. At first, we make this analysis for a single wire, then for a small wire array and finally we model and calculate very large arrays.

### 4.2. 2D Model for Single Wire

Figure 7 shows the amplification factor of a single permalloy wire with a diameter of 200 nm, length of 20 μm, and relative permeability *μ*_r_ = 500 inside a 20 μm long pick-up coil as a function of the coil diameter d. As the system is rotationally symmetrical, the calculation was made by 2D FEM.

The apparent permeability of this wire was also calculated by 2D FEM as *μ*_a_ = 361. Figure 7 shows that high values of the amplification factor can be achieved only when thin coil is fabricated tightly around the magnetic nanowire core. Even though, the achievable value of amplification factor is only 230, which is significantly lower value than the apparent permeability. When increasing the coil diameter, the amplification factor decreases rapidly; for 500 nm internal coil diameter and 50 nm coil thickness, the amplification factor calculated by FEM is only 50, while using Equation (4) the expected value would be a = 81. The reason of this behavior is that the magnetic flux density B around the magnetic wire core is weaker than the measured homogenous B_0_. The profile of B in the wire midplane is shown in Figure 8. When going from the wire center in radial direction, magnetic flux density B steeply drops upon crossing the boundary of the high-permeability core and air. The magnified part outside the wire shows that the field in the wire vicinity is weaker because the field lines are concentrated in the high-permeability region and this shielding effect is decreasing with distance.

### 4.3. 3D Model for Small Wire Array

Figure 9 shows apparent permeability and amplification factor as a function of wire array density Aw/Aair, of a small array of permalloy wires with a diameter of 200 nm, length of 20 μm, and relative permeability *μ*_r_ = 500. The wire array diameter is D = 20 µm, and the single-turn pickup coil has internal diameter of 22 µm, length of 20 μm and thickness of 1 μm or 50 nm. The apparent is decreasing with decreasing wire distance due to increasing magnetostatic coupling. For very small density the coupling is minimum and apparent permeability is approaching its maximum value of *μ*_a_ = 361 for single wire. The minimum value of *μ*_a_ = 4 is reached for 100% density, i.e., for solid permalloy cylinder with diameter of 20 μm, and length of 20 μm (Figure 9a). If we calculate the amplification factor using the simplified Equation (4), we obtain maximum amplification for 12% wire density (500 nm wire pitch). However, more accurate results obtained by FEM modelling show monotonous increase of the amplification factor with array wire density. The maximum value for both calculation methods is a = 3 for solid cylinder. The amplification factor only slightly depends on the coil thickness (Figure 9b).

### 4.4. Equivalent 2D Model for Large Wire Arrays

For the FEM simulations of large wire arrays, we proposed a simplified 2D equivalent model based on hollow cylinders. We also verified this model on small wire arrays of up to 90 wires [37] by comparison with 3D simulation and by measurement.

In the present paper, we extend these simulations to the nanowire arrays described in Section 2. As these arrays contain millions of wires, 3D FEM is impossible due to the computational complexity. Although we use symmetry to reduce the problem, even for 2521 wires in an array the number of elements is already 1.5 million and the computational time on a conventional PC (i7, 3.4 GHz, 8 cores with 32.0 GB RAM) reaches 43 min. Our 2D computational model attempts to overcome this problem-computation time for the same task was only 7 s.

Figure 10a shows a model of part of a hexagonal array of nanowires with diameter D_w_, length L, and distance d_w_. Figure 10b shows a similar square lattice model.

Figure 11 shows the equivalent 2D model that replaces wires by hollow cylinders. The height of the hollow cylinders is the same as the height of the wires. The mean radius of the hollow cylinders, *R_c_*, is calculated according to Equations (1) and (2), which are based on the assumption that the circle with radius *R_c_* has the same area as the red color hexagon/square shown in Figure 11a,b. The red color lines connect the centers of the wires into a single hexagonal/square “shell” of wires. The thickness, *t_c_*, of the hollow cylinders is calculated based on the assumption that the volume of the hollow cylinder must be equal to the volume of the wires that belong to the same shell.
(5)$$\begin{array}{c} R_{c}=\sqrt{3\cdot R_{h-c}\cdot R_{h-i}/\pi }=\sqrt{3\cdot \sqrt{3}/2/\pi }\cdot R_{h-c},\;R_{h-i}=\sqrt{3}/2R_{h-c}\;,\;\\ d_{c}=\sqrt{3\cdot \sqrt{3}/2/\pi }\cdot d_{w},\;t_{c}=3D_{w}^{2}/4/d_{c} \end{array}$$
(6)$$\begin{array}{c} R_{c}=\sqrt{4\cdot R_{r-i}\cdot R_{r-i}/\pi }=\sqrt{2/\pi }R_{r-c},R_{r-i}=\sqrt{2}/2R_{r-c},\\ d_{c}=\sqrt{2/\pi }d_{w},\;t_{c}=D_{w}^{2}/2/d_{c} \end{array}$$
where, *R_h_*_−*c*_ and *R_h_*_−*i*_ are the outer (circumference) radius of the hexagon and the inner radius of the hexagon, respectively. *R_r_*_−*c*_ and *R_r_*_−*i*_ are the outer (circumference) radius of the square and the inner radius of the square, respectively. The distance, *d_c_*, between the hollow cylinders is the same between all cylinders, as it is proportional to the distance of the wires, *d_w_*, as mentioned in Equations (1) and (2).

### 4.5. Verification of the 2D Model on Arrays of Thousands of Wires

In the first phase, we verified our 2D equivalent model by comparing it with a true 3D model. The calculation was performed for 20 μm wires 200 nm in diameter and for several distances between the wires. The wire lattice was either hexagonal (for 2791 wires) or square (for 2521 wires).

The calculated results are shown in Table 2. The maximum difference between the 3D model and the 2D model is 4%, showing that the 2D simplified equivalent model can be used with reasonable accuracy.

### 4.6. Using the 2D Model on Very Large Arrays

In the next phase, we modelled a 1 mm diameter membrane with embedded 200 nm wires (equivalent to Sample 1) by using the 2D-equivalent model. The wire length was again 20 µm. These arrays already contain from 200,000 up to 9 million wires, so that 3D FEM cannot be performed. In order to observe the effect of the material permeability and the geometry of the lattice, we performed simulations for hexagonal and square arrays and for relative permeability of 100, 500 and 1000. The simulation results are shown in Figure 12. For small distances, the apparent permeability depends neither on material permeability nor on lattice geometry, supposing that the permeability is 100 or more. For the minimum distance of *d_w_* = 275 nm, the apparent permeability is 2.25 for a hexagonal lattice and 2.6 for a square lattice. For the permeability 3.7 measured on Sample 1 (Table 1) the corresponding distance between wires is *d_w_* = 350 nm, which is only slightly lower value than the mean distance estimated from micrographs (500 nm).

As we already mentioned in Section 4.1., the multiwire induction sensor sensitivity depends not only on the apparent permeability of the core, but also on the coil geometry. The first approximation of the amplification factor *a*, calculated by using Equation (4), is shown in Figure 13. The values of the apparent permeability were calculated by FEM as a function of wire distance for a constant array diameter.

We have also used 2D equivalent FEM to calculate the amplification factor according to the definition in Equation (3). The resulting values are plotted in Figure 14 for the same parameters as in Figure 13. These results show that Equation (4) cannot be used for large arrays, as it gives unrealistic results. The estimates based on calculated flux are more precise.

Even though the apparent permeability of less dense large wire arrays can realistically reach the value of 80, the overall amplification factor of very large arrays is very small. This shows the non-intuitive result, that the sensors based on magnetic wires should have a diameter of the wire array lower than the wire length. This rule is applicable both for single-wire and multiwire cores. Several isles of such sub-arrays with their own pickup coils serially interconnected can be used in order to increase the sensitivity, but there should be distances reducing their magnetostatic coupling.

## 5. Conclusions

While soft magnetic wires of millimeter and micrometer size are successfully used in magnetic sensors, nanowire arrays have been developed mainly for magnetic storage applications and as microwave materials. This study has aimed to analyze possible applications of microwires and nanowires as sensor cores or field concentrators.

We have shown that magnetically soft nanowire arrays can be fabricated by controlling their shape anisotropy. The achieved coercivity value was as low as 4 kA/m. Future work will be targeted on further lowering the coercivity by applying a magnetic field during electrodeposition. Magnetic softening can also be accomplished by successive field annealing. High-temperature annealing may be possible after replacing the polymer membrane by an alumina membrane.

FEM magnetic modelling is essential for the design of future sensors. We have introduced a simplified 2D equivalent model, which allows the modelling of large arrays. The 2D model has been verified by comparison to full 3D model up to 2000 wires. According to simulations, for small pitch and very large arrays, demagnetization caused by magnetostatic coupling reduces the apparent permeability to small values regardless of the permeability of the material. The simulation results fit the values of *μ*_a_ = 3 measured by SQUID magnetometer for an array of millions of 200 nm diameter wires.

We have also shown that the widely used apparent permeability can be employed to characterize single-core sensors with slim coil, but it gives misleading results for multicore sensors. Therefore, we use the amplification factor, which is directly related to the sensitivity of the induction sensor based on a given combination of core and coil. We have shown that only for single wire and very small wire arrays the amplification factor can be estimated by the popular simplified formula. For larger arrays, the simplified formula gives unrealistic results (false maximum appears even for 91 wires) and the amplification factor should be calculated by FEM. While for a small array inside a 20 µm diameter microcoil the amplification factor is 3.2, for a 1 mm diameter coil the correct amplification factor drops down to 1.02. This indicates that another key challenge for the future development of nanowire-based magnetic sensors is to fabricate microcoils small enough to be able to efficiently capture the signal from nanowire array.

## Figures and Tables

**Figure 1 sensors-21-00003-f001:**
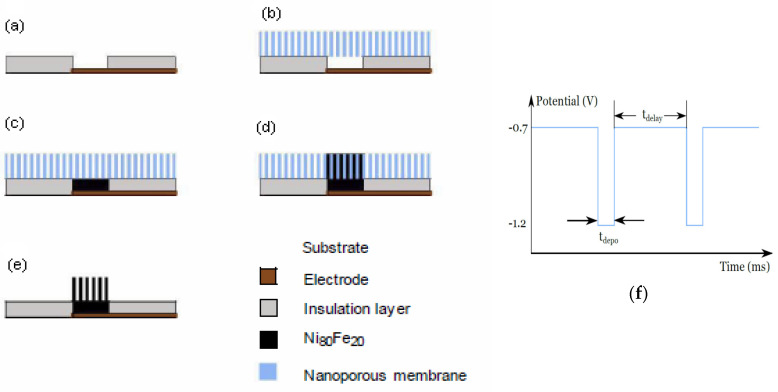
Fabrication of nickel-iron nanowires using nanoporous membranes: (**a**) insulation layer (photoresist) on top of sputtered Cu electrode, (**b**) membrane placed, (**c**) Filling of the holes by electordeposition, (**d**) growing nanowires, (**e**) optional dissolving of the membrane not used in this study, (**f**) waveform used for electrodeposition.

**Figure 2 sensors-21-00003-f002:**
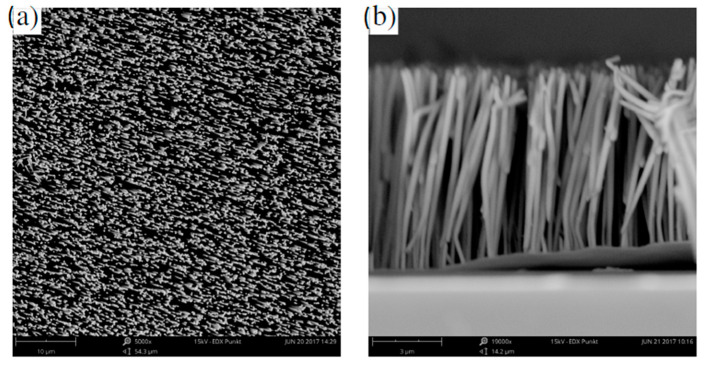
SEM micrographs of a NiFe wire array with a wire diameter of 200 nm, length of 20 µm, and an aspect ratio of 100. (**a**) top view and (**b**) lateral view.

**Figure 3 sensors-21-00003-f003:**
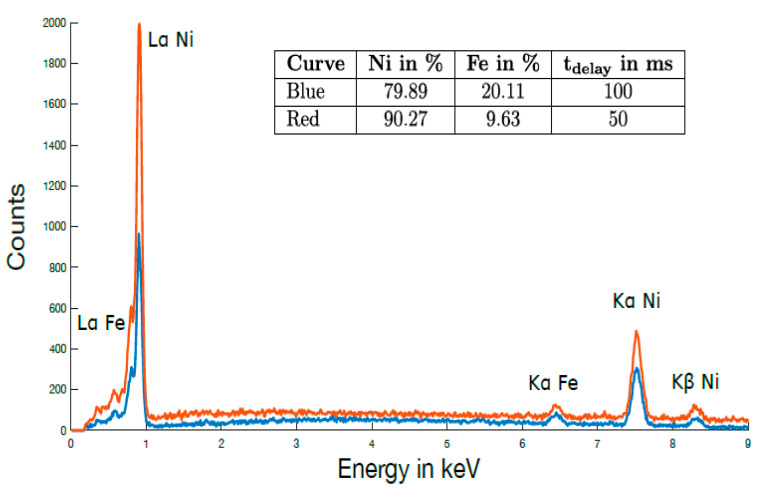
Energy dispersive X-ray spectroscopy (EDX) spectrum of the Ni80Fe20 and Ni90Fe10 nanowire arrays. Differences in the composition were achieved by using different delay times in the pulsed deposition process.

**Figure 4 sensors-21-00003-f004:**
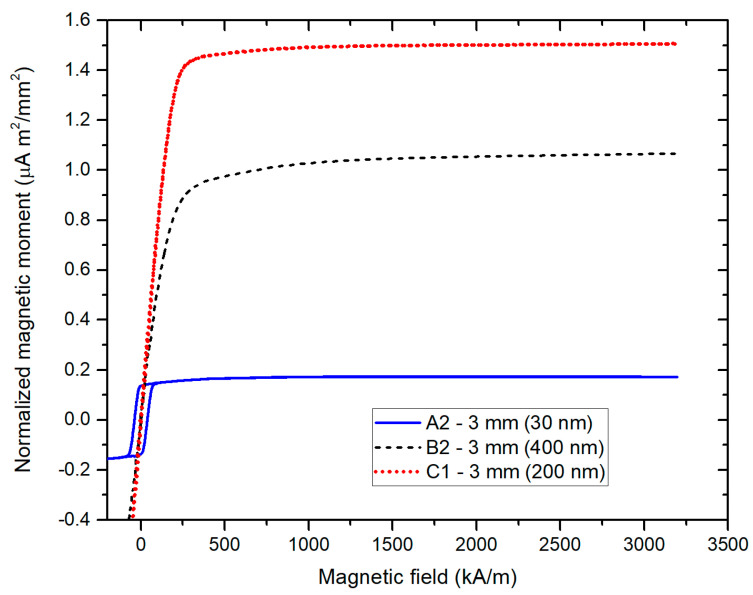
High-field hysteresis loops of the arrays of 20 µm long wires with different diameters.

**Figure 5 sensors-21-00003-f005:**
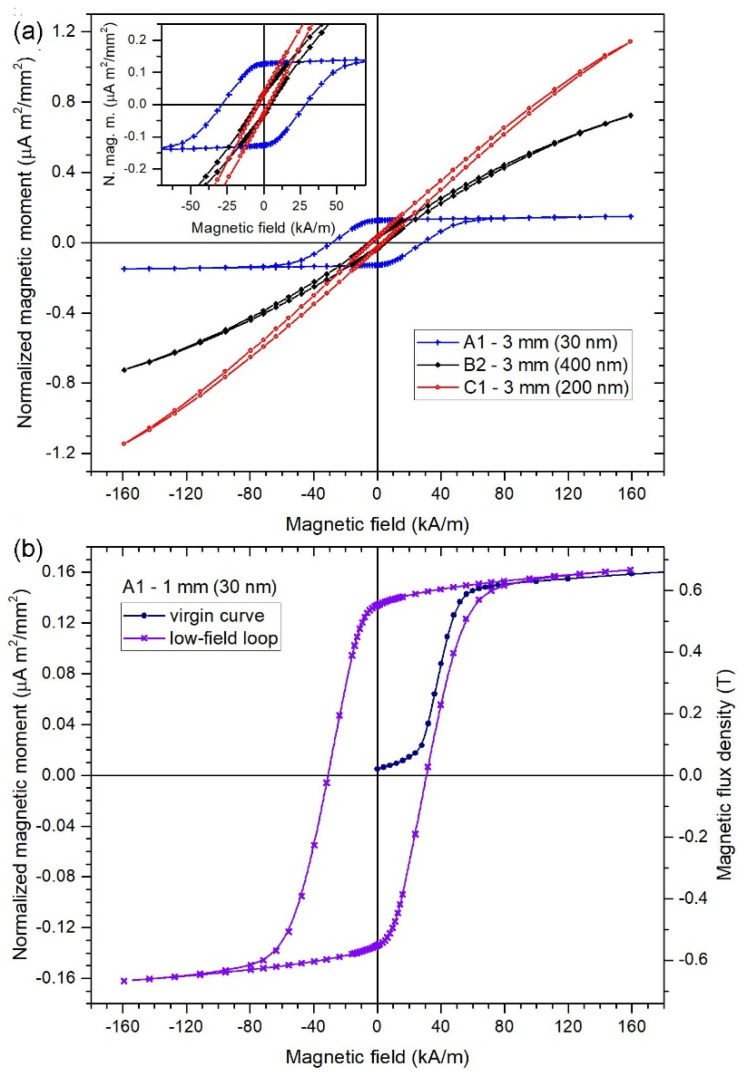
Low-field hysteresis loops of (**a**) 3 mm diameter circular arrays of Permalloy wires with different diameters. The wire length is always 20 µm. The inset shows the low-field details of the same loops. (**b**) 1 mm diameter array of 30 nm diameter wires also showing the virgin curve.

**Figure 6 sensors-21-00003-f006:**
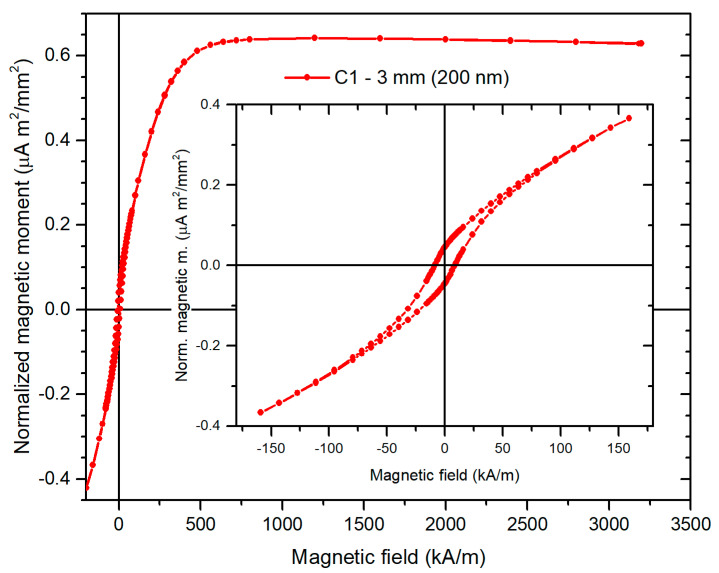
Low-field hysteresis loop of the array of 200 nm diameter 20 μm long permalloy wires measured in the direction perpendicular to wires.

**Figure 7 sensors-21-00003-f007:**
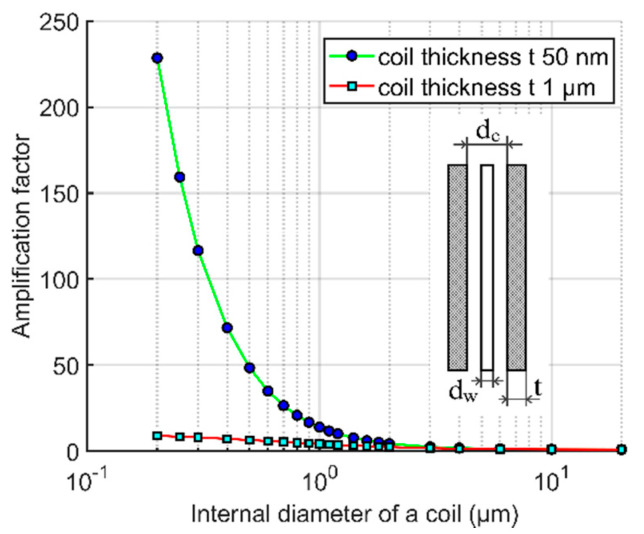
Amplification factor of a single 20 μm long permalloy wire with a diameter of 200 nm inside a 20 μm long pick-up coil with two different thicknesses as a function of the coil diameter d (FEM simulation). The wire material relative permeability is *μ*_r_ = 500 and apparent permeability *μ*_a_ = 361.

**Figure 8 sensors-21-00003-f008:**
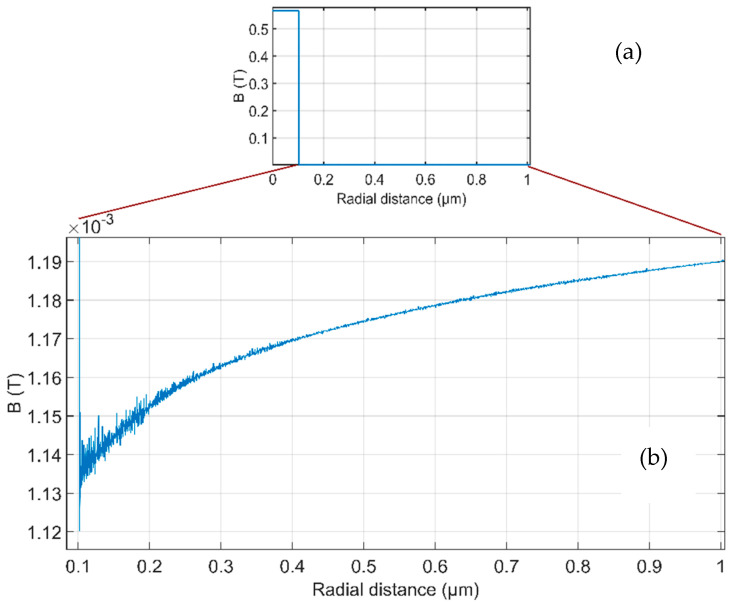
Magnetic flux density B in the midplane of a 200 nm diameter wire as a function of radial distance from the wire center: (**a**) inside and outside the wire (**b**) the outside part enlarged to show decrease in B close to the wire surface.

**Figure 9 sensors-21-00003-f009:**
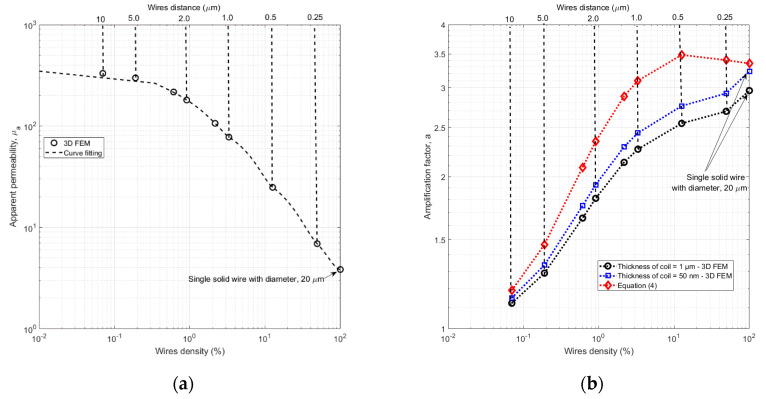
Apparent permeability (**a**) and amplification factor (**b**) of a small wire array as a function of wire density (FEM simulation). The array diameter D = 20 µm is the same as the wire length.

**Figure 10 sensors-21-00003-f010:**
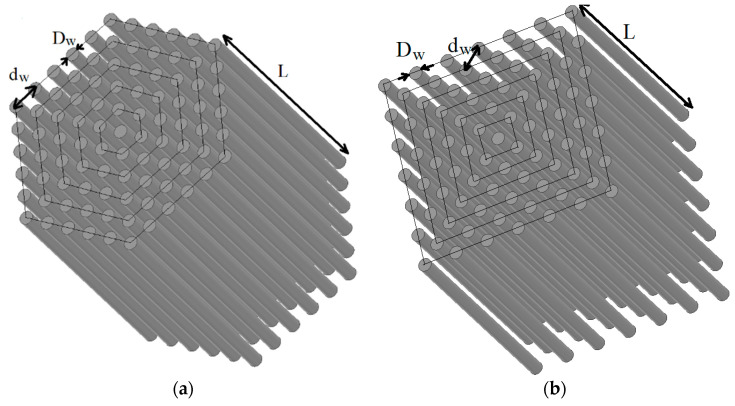
Part of the model of the wire array (**a**) with a hexagonal arrangement and (**b**) with a square arrangement.

**Figure 11 sensors-21-00003-f011:**
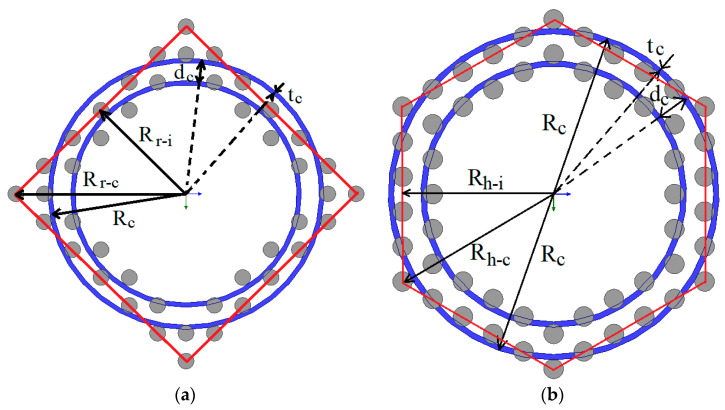
Equivalent hollow cylinders (**a**) for a hexagonal arrangement and (**b**) for a square arrangement of the wires.

**Figure 12 sensors-21-00003-f012:**
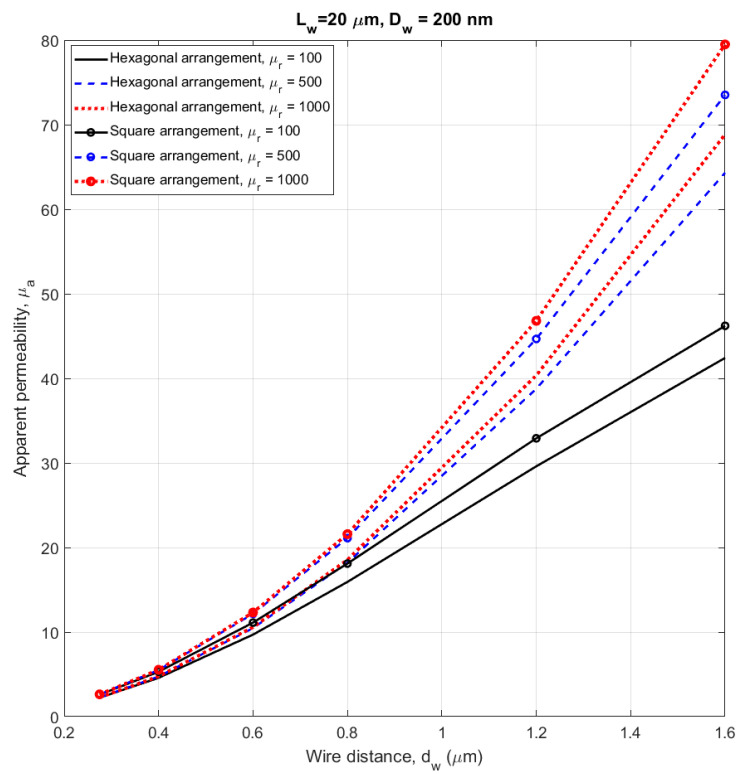
Apparent permeability versus wire distance with hexagonal and square arrangements–Membrane diameter, D_m_ = 1 mm. Calculated by FEM using the 2D-equivalent model.

**Figure 13 sensors-21-00003-f013:**
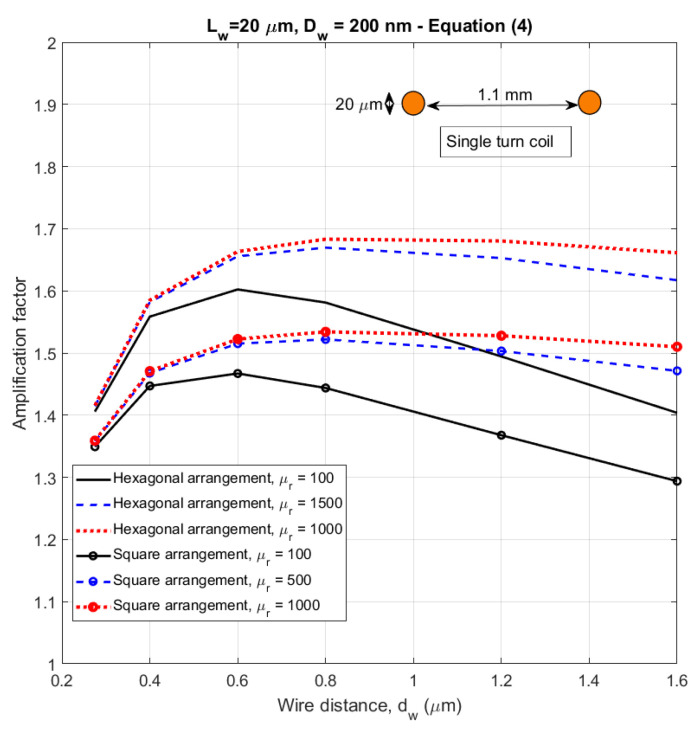
Amplification factor of a 1 mm diameter array of nanowires as a function of wire density–calculated by the approximate Equation (4).

**Figure 14 sensors-21-00003-f014:**
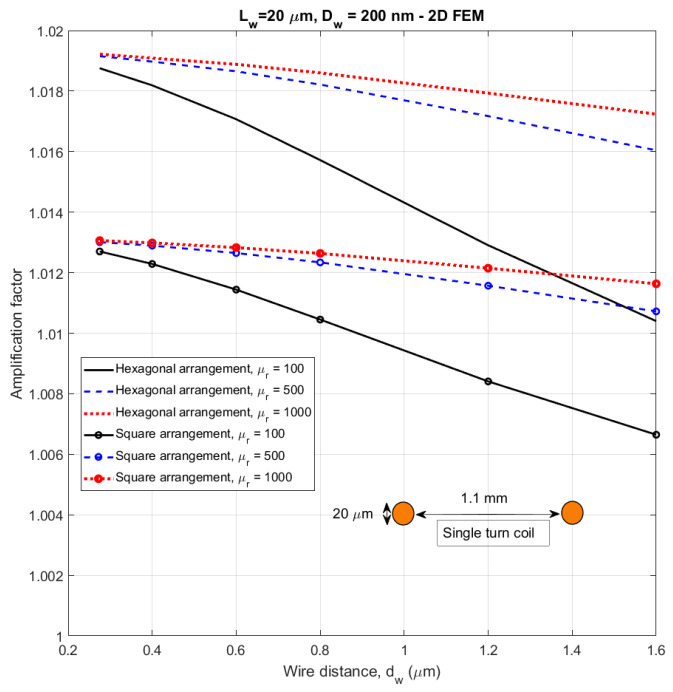
Amplification factor of a 1 mm diameter array of nanowires as a function of wire density–calculated by FEM.

**Table 1 sensors-21-00003-t001:** Measured coercivity and calculated permeability values of arrays of 20 μm long permalloy nanowires.

#	Sample	Hc [kA/m]	*μ* _a_
1	A1-1mm 30 nm	31	20
2	A1-3mm 30 nm	29	18
3	A2-3mm 30 nm	39	16
4	A4-1mm 30 nm	31	14
5	B2-1mm 400 nm	3.9	3.3
6	B2-3mm 400 nm	4.5	3.4
7	C1-1mm 200 nm	4.6	3.7
8	C1-3mm 200 nm	2.6	3.3

**Table 2 sensors-21-00003-t002:** Apparent Permeability Comparison between 3D and 2D.

Case-*µ*_r_ = 500 *D_w_* = 200 nm, *L_w_* = 20 µm	*µ*_a_ 3D	*µ*_a_ 2D	*Rel. Diff. (%)*
2791 wires–Hexagon, *d**_w_* = 1.6 µm	85.75	88.39	3.1
2791 wires–Hexagon, *d**_w_* = 1.2 µm	59.40	60.24	1.4
2791 wires–Hexagon, *d**_w_* = 0.8 µm	34.91	34.94	0.1
2791 wires–Hexagon, *d**_w_* = 0.6 µm	24.13	23.97	−0.7
2791 wires–Hexagon, *d**_w_* = 0.4 µm	14.72	14.51	−1.4
2791 wires–Hexagon, *d**_w_* = 0.275 µm	9.68	9.49	−2.0
2521 wires–Square, *d**_w_* = 1.6 µm	97.06	98.98	2.0
2521 wires–Square, *d**_w_* = 1.2 µm	67.13	67.90	1.1
2521 wires–Square, *d**_w_* = 0.8 µm	39.84	39.52	−0.8
2521 wires–Square, *d**_w_* = 0.6 µm	27.83	27.13	−2.5
2521 wires–Square, *d**_w_* = 0.4 µm	16.92	16.40	−3.1
2521 wires–Square, *d**_w_* = 0.275 µm	11.13	10.70	−3.9

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
