# Peer review of "Modelling and Measurement of Magnetically Soft Nanowire Arrays for Sensor Applications"

_sensors, 2020, doi:10.3390/s21010003_

Round 1
Reviewer 1 Report
Please check the numbering of your figures
Page 4: What kind of anode you use for your electroplating process? inert PT-anode or soluble anode
Page 6 and 7: Why is there no hysteresis at 200nm and 400nm? Actually the hysteresis gets smaller the smaller the structure.
Page 7: Py is Permalloy?
Page 7 line 188: it should be 30-40 kA/m
Page 9 line 228: Please delete "initial calculations"
Page 9 Please check the font
Page 10 Fig. 8: Please improve the quality of the figure. It is not readable in the pdf
Page 11 line 303: Compare to the 3D modelling what is the time for your model.
Page 14: Please check the font
Reviewer 2 Report
The authors have been studying this topic for a while and this paper describes their most recent results. It is an excellent paper. I am curious to know if their numerical results for the variation of the anisotropy with the geometrical form of the fibers are consistent with simple analytical models, such as the Stoner-Wohlfarth model with the anisotropy coming from shape anisotropy. Or does the anisotropy come from a different source?
The authors should respond to the following:
- Figures 2, 3, and 4 are mislabeled and are presented as Fig. 3, Fig. 2 and Fig. 2. Please correct the numbering.
- In line 252, the correct spelling is length
- Since the coil thickness is an important concept, please include a figure to show which dimension of the wire-coil system is the coil thickness.
- In line 268 the boundary condition on the magnetic field parallel to the wire surface is that the parallel component of the field must be continuous at the boundary. Why do the authors claim otherwise?
- The authors should elaborate more on the non intuitive result presented in lines 376-377. It seems that the tightly wound pickup coil will capture all of the magnetic flux in the sample whereas the larger coil will not. Doesn’t this indicate an error in the calculations?
The authors are publishing another paper this year,Reference 50, on the same topic. It is impossible to know how much overlap there is with this manuscript since that reference is not yet available.
Reviewer 3 Report
Some small technical improvements could be recommended:
The authors should consider renumbering of the figures. Now Figure 3 precedes Figure 2. Also, there are two figures numbered Figure 2.
There should be a proper axis title for the y-axis in Figures 4, 5 and 6. Please pay attention especially to Fig. 5b.
It is not recommended to use both comma and full stop for decimal point.
